# An invisible caregiver for visibly older parents: Experiences of (young) adults shared as comments to newspaper articles on advanced age parenthood

**Kato Verghote** *, **Priya Satalkar-Götz**, **Guido Pennings**, **Veerle Provoost**

Department of Philosophy and Moral Sciences, Bioethics Institute Ghent, Ghent University, Ghent, Belgium

☉ These authors contributed equally to this work.
* Kato.Verghote@UGent.be

## Abstract

At the centre of the debate on advanced age parenthood are concerns for the offspring's well-being. In the few empirical studies available, researchers found that children born to older parents show similar or better cognitive, behavioural and psychosocial outcomes compared to children born to younger parents. Most of these studies examining the children's perspective are quantitative. This study qualitatively examined the experiences presented by (young) adults who identified as born to older parents offered in response to a selection of newspaper articles on the topic. Performing inductive thematic analysis, we found that positive experiences were often presented as a way to contradict prejudices about advanced age parenthood. Other comments described the visual representation of the older parents' age as an attribute that created difference and, in some cases, social distance from peers and the outside world in general. Central to the negative experiences was a contrast between the visibility of being a child of older parents and the invisibility of caring for them. Moreover, in a majority of the latter comments, the commenters' caregiving experiences combined with the social network's notable lack of understanding and support regarding the caregiving responsibilities were described as adversely affecting their lives. These findings provide more insight into the experiences of (young) adults born to advanced age parents and their potential needs.

## Introduction

In most western societies, the age at which mothers and fathers first enter parenthood is on the rise, with successive births only increasing the average age of parenthood [1, 2]. This general trend has been framed in the literature as 'late parenthood', 'postponed parenthood', or 'delayed parenthood' although it must be noted that there is a general lack of consensus about what constitutes 'older' parenthood. Researchers have been using varying age cut-offs [3], where it is typical to refer to women between the ages of 30 and 40 as 'older' [4–6], and older

**Data Availability Statement:** All relevant data are within the paper and its Supporting Information files.

**Funding:** KV, GP & VP received funding for this work. This study is funded by FWO (Fonds Wetenschappelijk Onderzoek) (FWO. OPR.2021.0001.01). URL to funder website: https://www.fwo.be/en/ The funders had no role in study design, data collection and analysis, decision to publish, or preparation of the manuscript.

**Competing interests:** The authors have declared that no competing interests exist.

men are categorised as such in their 40's and 50's [7]. Multiple factors are cited as influencing the postponement of motherhood, which include: the couple's health, financial stability, home environment, structural factors like a longer average period spent in education and a difficulty to balance a time-intensive profession with family life, being single or relationship uncertainty and breakdown, and individual emotional preparedness [8, 9]. Similar factors have been found to be associated with the postponement of fatherhood [10]. In addition, the lack of supportive family policies seems to contribute to this 'postponing' trend [11].

Advanced age parenthood (AAPd) seemingly raises several concerns for future children. These concerns, although intuitively appealing, often lack the attendant empirical findings to support the degree of attention they receive in scholarly publications. First, the literature mentions strictly medical concerns, such as an increased risk of miscarriage and chromosomal aneuploidy for children born to 'older' women. The risks are not exclusive to older women, as children born to older men are at an increased risk for neuropsychiatric disorders and certain chromosomal abnormalities [12]. In addition, several scholars point out the poor psychosocial outcomes for the offspring of advanced age parents (AAPs). A potential source of such poor psychosocial outcomes is a lack of adequate parenting skills of older parents who would not be able to fulfil their parental duties [13]. Another critical concern for the children of AAPs is the increased risk of losing a parent at a young age which brings severe grief, stress and loss of material and emotional support [14, 15]. Moreover, the increased risk of degenerative illness in the parents is often stated as a source of poor psychosocial outcomes for these children [16]. The underlying concern is that the children will be younger as their parents start experiencing more serious health problems, forcing them to "face the burden of caring for their parents" [17] earlier in life.

The above concerns, however, have no clear evidence base. Overall, a number of quantitative studies indicate that children born to older mothers show similar or better cognitive, behavioural and psychosocial outcomes compared to children born to younger mothers. Barclay and Myrskylä [18] found that Swedish persons born to older mothers performed better on standardised tests and had a higher degree of educational attainment than their peers who were birthed by younger mothers. Boivin and colleagues [5] found similar correlations when measuring wellbeing (among 4–11 years old children) across British families of younger and older mothers. One research team found slightly more negative cognitive and behavioural outcomes for children of advanced age mothers compared to children of younger mothers but later concluded that this trend reversed since the beginning of this century [19, 20]. This was explained by the fact that more recent cohorts with children of older parents are often situated within socioeconomically advantaged families. All studies listed here exclusively examined the outcomes of children born to older mothers, and not to older fathers. Moreover, they focused almost entirely on minors, whereas adult children born to older parents have rarely been studied. We were only able to find two studies of qualitative accounts of (adult) children born to older parents [21, 22]. Their main findings included that (adult) children born to older parents (i) felt different from peers with younger parents, (ii) regretted to be part of a small (extended) family, having no siblings or living grandparents, or feeling estranged from much older (half) siblings, (iii) feared premature parental death, (iv) reported having more caregiving responsibilities than their peers, and (v) identified both advantages (e.g., greater wisdom, patience and financial resources compared to younger parents) and disadvantages (e.g., less affinity to the young child's or teenager's social and emotional reality compared to younger parents) to their parents engaging in late parenting. However, both studies date back a quarter of a century and are more journalistic in nature. With this study we aim to provide an updated explorative analysis of the experiences of (young) adults born to AAPs and to foster societal understanding of what it means to these people to be a caregiver for their parents at a relatively young age.

As it has become increasingly common to discuss parenting and parenthood in general on social media platforms, internet forums, and comment sections on other types of websites (e.g., blogs, online newspaper articles) over the last few decades [23], so have discussions of AAPs on these mediums. Online newspaper articles often spark personal responses providing a rich source of data. In this paper, we examine what experiences are offered by (adult) children with AAPs in the comment section of newspaper articles published in The Guardian (TG) and The Daily Mail Online (DMO). We studied data stemming from people with first-hand experiences with AAPs, and who expressed their experiences in an unsolicited way. In doing so, we aimed to identify what they consider relevant for the debate on AAPd, and we hope to give voice to these people and initiate further research.

## Materials and methods

### Study site

A search in multiple British online newspapers (including The Daily Telegraph, The Times, The Independent, and The Sun) revealed that TG and DMO provided the largest number of articles on the subject of older parenthood. The elaborate and personal comments on many of these online newspaper articles in TG and DMO provided sufficiently rich material suitable for a qualitative analysis.

TG has been identified as leaning left-wing [24] and targets a well-educated, relatively young audience [25], whereas DMO has been identified as political right [26] and targets lower to middle-class readers [27]. Moreover, both newspapers are known for their wide and international readership. In both newspapers, commenting on articles is possible only through registration whereas accessing these comments is possible for every Internet user without requirements to register.

### Data collection

We searched TG and DMO for articles on the subject of older parenthood using the combined terms 'older parents' or 'older parenthood'. Articles were selected based on the following inclusion criteria: they (i) focussed on lay people drawing from their personal experience (rather than on research findings or expert opinions), (ii) had 100+ comments (including comments on comments), and (iii) had been published no longer than ten years ago. We ultimately selected six online articles, three from each newspaper.

For this study, we used passive data collection [28]: we did not contribute to or manipulate the online discussions. Comments that did not abide by the community standards of the online newspaper were removed by a moderator and were inaccessible to any Internet user. Of all accessible comments (1.248) we created a data subset of comments in which it was clear that the commenter was an individual born to (an) older parent(s). Rather than precisely defining 'older parenthood' by taking a particular age cut-off as a benchmark, we used the commenters' descriptions and experiences of having or living in a family with older parents. Commenters made this clear by either explicitly stating their parents' age at the time of the commenter's birth (this age ranged from 38 to 69 years old), or by describing that they were born to older parents in more general terms (e.g., "I had older parents", or "My parents were older when I was born"). The final data subset for this analysis comprised of 151 comments of 20.144 words in total. The richness and depth, rather than the quantity of the data, were assessed by all authors in light of the qualitative analysis. This is consistent with other research drawing on online data [29, 30].

The fact that commenters gave their unsolicited responses to these newspaper articles allowed us to gain insight into the experiences and concerns they themselves selected to

bring into the public sphere and comments they regarded as relevant for a societal debate about AAPd. It is in this way that the comments were studied, rather than as extensive accounts of experience such as those collected in interview studies. The unit of our analysis consists of newspaper comments rather than the individuals behind the comments (given that the study design was not aimed at gaining an in-depth insight into these individuals). Therefore, in the results below, we make statements about comments and not about the commenters.

## Data analysis

We analysed the comments using reflexive inductive thematic analysis [31] facilitated by NVivo (v.1.6.1), a qualitative software programme. Inductive thematic analysis was used as we aimed to identify recurring "patterns of meaning" [32] across our dataset of collected comments. The reasons for choosing this method of analysis include high flexibility of the researchers, the opportunity to thoroughly examine data that stem from a variety of sources, and the ability to identify both differences and similarities across data sections. Moreover, choosing for a data-driven analysis allowed us to develop new insights one might not have anticipated using a strictly deductive approach. The first author conducted the first round of coding, and drafted a list of codes and potential themes which were subsequently analysed alongside a selection of the pseudonymised data by two additional co-authors. Auditing meetings were organised in collaboration with all three co-authors [33]. The auditors' role was to critically challenge and discuss provisional theme structures and to help refine potential themes. The auditors contributed to the analysis from their respective disciplinary backgrounds (i.e., medicine, public health, medical anthropology, gerontology and philosophy), thereby often challenging interpretations and/or suggesting alternative interpretations which added to the depth and rigour of the analysis.

## Ethics

Different strategies have been presented to conduct qualitative research using online data [34, 35]. Following Snee [36], we decided to view the commenters in this study as authors, individually producing and publishing public online content. As Snee [36] indicates, if we treat commenters as authors then their authorship should be recognised in the form of a source reference. However, we had to balance the commenters' need for authorship against their need for privacy, because we cannot assume that all online users are aware of or fully understand the extensive consequences of publishing online comments. By default, we should assume that online users do not want to be recognised as the author of a particular comment in scientific research. Therefore, we minimised the reference to individual commenters and took the following two precautions to protect their privacy and prevent them from being too easily retraced by the use of online search tools [37]. First, we removed commenters' original pseudonyms or real-life names. Second, personal information (such as name, age, residence) that could (in)directly lead to the identification of the commenter was removed or modified in a way that did not change the meaning of the original comment. We referred to age periods such as 'teens' or 'mid-50's' instead of to commenters' exact ages except in a limited number of comments where this constituted an important interpretative element. Only the first author had access to the raw dataset. Given that data were publicly available, informed consent was not directly obtained from the commenters included in this study. Approval was received from the Ethics Committee of the Faculty of Arts and Philosophy, Ghent University (Ref. 2021–12).

## Results

Overall, there was a comparable number of comments using more positive (N = 74) and more negative (N = 77) wording in describing their experiences with AAPd. Whereas the positive experiences were usually articulated in more compact and general terms, the more negative comments were often lengthy, detailed, and emotionally loaded, possibly indicating a need to be heard and understood. In what follows, we discuss three themes. The first one (“*Addressing prejudice by pointing out the positive*”) relates to the experiences presented by the commenters as counterexamples to prevalent prejudices about parenting later in life. In these comments, the commenters were pointing out the positive in their experiences with AAPs. The second (“*Being a child of visibly older parents*”) and third (“*Doing the caring in an invisible sort of way*”) theme describe the experiences presented by commenters as a way of raising concerns and warning others in relation to AAPd. Apart from experiences of being visible to the outside world as a child of AAPs, these comments also highlighted the invisibility of taking up an untimely caregiver role for AAPs.

### Addressing prejudice by pointing out the positive

Many of the commenters’ positive experiences were written in a particular way; as an attempt to convince others of the benefits of being born to older parents, and in doing so, occasionally overstressing the positives. Moreover, the personal experiences were used as counterexamples to address two types of prevailing prejudices regarding AAPd: the idea that children of older parents miss out on important things during and beyond childhood and the general depiction of AAPs as looking old, suffering from bad health, being inactive and/or needy. Below, we describe two subthemes (“*Not missing out*” and “*Still going strong*”) in which we look into these two types of prejudices respectively.

### Not missing out

These commenters emphasised that their parents provided them with a “*great*” and “*wonderful*” childhood despite their parents’ advanced age. They depicted the older parent as a good parent not in spite of but because of his or her older age. We found several commenters to employ a comparison strategy in order to highlight their older parents’ advantageous characteristics compared to younger parents, such as life experience, availability, and a genuine commitment to parenting, which in the commenters’ eyes make a good parent:

> I don’t think (although maybe because I don’t have any other experience) that being an older parent is a detriment, if anything I feel that it’s a good thing. Both of my parents retired whilst I was at school so we spent more time together which is something that a lot of my friends didn’t experience due to their parents working.

> My father was in his mid-40s when he had me [. . .] but a positive to this is that he had a wealth of life experience and as was able to retire around fifty. I had a full time dad which was excellent.

> My mother’s age has never once had a negative impact on my life, quite the contrary as she has so much wisdom and life experience that she shares with me every day.

Considering their older parents’ advantageous characteristics, many of these commenters felt “*lucky*” to have had such a parent, and some said they would not have preferred it any other way.

Several commenters explicitly disentangled age from how they assessed their parents. They attributed their parent's good parenting to their personal characteristics, attitude, and undertakings–which, as one commenter put it, "*age has nothing to do with*".

## Still going strong

Many commenters who shared an overall positive experience with their AAPs tended to stress how their parents were "*still going strong*". They anticipated a typical prejudice against older parents as being inactive and needy. To prove the inaccuracy of this prejudice, these commenters emphasised their parents' young appearance, high fitness level, good health and/or independent living. Whilst appearance, energy, health and (in)dependent living and functioning were also topics present in the articles they commented on, the commenters chose these elements to address using their own experiences:

> When I was born, my father was in his early-50s, and my mum was in her early-40s. They were both younger in outlook (mum) and appearance (both) than many of my friends' much younger parents. I never felt embarrassed about them.

> They were always healthy and energetic and still pretty much are. No problems. Not everyone has health problems.

Whereas the last two comments demonstrate the absence of a caregiver role for one's parents, below we describe how taking on such a role was at the heart of many of the more negative experiences with AAPs.

## Being a child of visibly older parents

Commenters often recalled childhood memories of being aware that they were visibly recognizable to the outside world as a child of AAPs. In the first subtheme ("*Sensing difference*") below, we identified how commenters described having AAPs as a feature that made them feel different from peers. At the same time commenters recalled being judged by outsiders for having AAPs, and described to have managed such social judgement in different ways, as we present in the second subtheme ("*Dealing with social judgement*").

## Sensing difference

Throughout their childhood, some commenters said to have come across social situations in which their awareness of having an older parent was triggered, mostly by people outside the family (e.g., peers, teachers):

> I remember once when I was at primary school and my father came to pick me up from dance lessons, my teacher mistook him for my grandfather. That moment has always stuck in my memory, as up to that point I had never noticed any difference.

> I think it hit home that I had a much older dad when our history teacher asked who had a dad over 50 when they were born.

These commenters recalled a key moment in their childhood which made them reflect on how outsiders perceived their older parents. These outsiders took physical appearance, mobility, and health as a basis to determine the parents' age, and subsequently to determine their appropriateness or suitability to be a parent. Outsiders "*laughing*" or giving "*funny looks*" as a

response to observing the older parent indicated that something was wrong or inappropriate, as these commenters explained:

> I had a great childhood but my dad started to show signs of Parkinson when I attended secondary school. I remember other children laughing at him in the car when he had to wear a neck support because his head was shaking so much. He used to pick me up from school. I remember my mother's clothes always being more ahem mature than other mothers, small things you would notice that differentiated you from the rest.

> I remember the strange atmosphere and funny looks from my own childhood whenever I went anywhere with my dad.

Older age–and the visual representation of that–was therefore presented as an attribute that created difference and, in some cases, social distance from peers and from the outside world in general.

**Dealing with social judgement.** Most commenters who described to have been confronted with outsiders' judgements on their parents revealed to have felt embarrassed and ashamed. They thought these judgements made by the people in their social network were justified. They thus accepted others' perception of inappropriateness of AAPd, as they tacitly endured the judgements without questioning them:

> I was always uncomfortable when I saw them next to the other parents. One of my friends once asked 'who's the old woman' when she first saw my mother and I was mortified.

> I loved my parents but as I got older I felt embarrassed as they were in their 40s when they had me. I rarely wanted them to come to parents nights at school as all the other mums were young, in fact several times I was asked if my mum was my grandmother, pretty embarrassing for a teenager.

The latter quote also indicates that this commenter tried to avoid outsiders' judgement altogether by shunning to be seen with their older mother in public spaces. Inherent to several similar quotes was a tendency to compare one's parents to those of peers. Commenters often perceived peers' younger parents as the norm of ideal parenthood, which in a few comments was accompanied with a desired image of a parent:

> Through my eyes as a young child I wanted a young, modern trendy and up-to-date mum like the other children.

Some commenters, however, reflected upon such outsider judgements later in their lives. Although they confirmed to have experienced negative comments on their parents, they tried to put these experiences into perspective. They reasoned this confrontation was not such a big deal to them after all and some even mentioned valuable life lessons learnt from this (initially negative) experience:

> I do remember being a bit embarrassed about having older parents when I was a child but I don't think it was any more of a problem than all the embarrassing things my friends thought their parents did. I was never more bothered by their age than my friends were about things like their parents' dancing or awful jokes.

> Yes I also had people ask about why I was going out with 'grandad' but it taught me to not care about others' opinions of my life.

## Doing the caring in an invisible sort of way

A significant part of the focus in the comments went to the commenters' experiences with being a caregiver for their older parent(s). In the first subtheme ("*Timing of caregiving*") we consider some commenters stating they took on a caregiver role during their childhood/adolescence, whereas most explained they cared for one or both of their parents during their (early) adulthood, taking on tasks ranging from managing the household to acting as the parent's "*full time health caregiver*". Whilst not going into much detail about what exactly their caregiver role entailed, the commenters underscored society's incomprehension of their caregiver role as presented in the second subtheme ("*Societal ignorance*"). Moreover, in the third subtheme ("*Emotional costs of caregiving*") we identified how the commenters described to feel about their caregiver role and what it meant to them in their personal lives.

**Timing of caregiving.**    One commenter who acted as a caregiver for their father during their teenage years expressed feeling a lack of parenting due to the family's focus on the father:

> Grieving for a living person with dementia is hard; it's harder when you yourself are a teenager, or 20-something, and really do need parenting.

> Would it have been easier to cope with that [caregiver role] at 22? It would have been hard, but I would have had a dad (and mum) to rely on in my teen years, and perhaps more faith in my own observations.

Moreover, this commenter pointed to experiences of parentification: they cared for their younger sibling, drove the parents around and checked the father's medical adherence. They described these tasks as "*intense responsibilities*" for a child at that age. Another commenter expressed similar experiences of parentification, stating they were treated as "*a substitute*" for an absent spouse or second parent, mainly providing emotional care to their older mother:

> My mum was in her early-40s when she had me, and brought me up on her own. I clearly remember one time we were skint and she was having a good moan about money, and she finished off saying something like 'It's so much better now you're older and I can talk to you about things like money problems.' I think I was about 9. I think sometimes single parents/one of a couple who has to care for the other must feel lonely and frustrated at not having an adult to consult and confide with, and their child becomes a substitute.

Commenters stating they cared for their older parent(s) during their adulthood shared different experiences, mainly depending on when exactly they started taking up this role. Some explained the care work started at a time when they were building a family of their own. Most of these commenters revealed they did not follow the same reproductive timing as their parents, and thus started a family earlier, at a younger age (i.e., in their 20s or 30s). This meant they had to take care of ageing parents and little/teenage children at the same time which they felt somewhat incompatible with a full-time job and "*normal*" family life. Commenters explicitly mentioned that their double caregiving role affected their health, their relational and/or professional life:

> All the time, I was holding down a full time job and raising my son, who was born when I was in my mid-20s. My mother finally died when I was in my early-40s and left my health in a precarious condition from the years of stress, not to mention it destroyed my marriage, due to the constant care she needed which my husband resented very much for so many years.

> If I thought it was a struggle to maintain a home, relationship, two kids and a decent full time job, but it became impossible when I became the person who drove them [my parents] to hospital appointments, did their shopping and got called in the night when one or other had fallen, felt unwell or wet the bed. My father died, my relationship of 20 years broke down and I gave up a modest career for a part-time job for the next four years until my mother followed him.

Several commenters stated they did not have a family of their own and attributed their current family situations (i.e., unmarried, or single, and childless) to their caregiver role and the impact it (has) had on their lives. These commenters explained it was precisely this time-consuming role that prevented them from pursuing higher education, a (more ambitious) career, a relationship or a family:

> I ended up at 18 years old having to give up any hope of university and nurse them [my parents] both.

> I have a good job in retail, but it's not a career, it's a job that pays the bills and allows the flexibility I need to care for my parents. I have been a carer since I was in my early teens; so, no career like my peers, and I most certainly do not have a partner or children of my own.

> I feel that I made an almost unconscious transition from child to carer, and my own life has been on hold. I have managed to get a degree, and I have held down a job, but I am reaching a stage where I may have to make a choice about whether to care for mum, or continue to work. That won't really be a choice for me. Work is work, but family is what matters. But I fear for the future.

**Societal ignorance.** Often, what seemed to make the caregiver role harder to bear was the lack of support. Many commenters revealed to have been confronted with a lack of understanding for their caregiver role in their social network (e.g., friends, colleagues) and in institutional structures (e.g., employers, the law):

> I also arguably experienced career setbacks caring for my mother at a young age, for neither my employer nor the law recognized that I needed time away to do this.

> It's also frustrating that people rarely understand that caring for parents and losing them is as much of a passage into adulthood and responsibility as having kids. [. . .] I am treated like someone who hasn't grown up and had to take on responsibilities beyond myself because I have no kids and my partner and I are unmarried.

Another commenter with a caregiver role as a minor described a similar experience: worries about their family situation were typically ignored or not taken seriously resulting in a complete absence of support:

> When I tried to articulate my concerns to *any* adult, and *any* point about *any* topic, it was dismissed as the moanings of a typical teenager. Which I took to heart, and decided that it was *my* judgement, and *my* powers of observation which were in error.

> I've been envious of people who have lost a parent when they were a child, teen, or 20-something, because you see other people in the community step up to help. My parents are still "there", so no one helped.

One commenter reasoned that this societal incomprehension could largely be attributed to the prevailing norm that people "*expect to be in their mid-50s before they have to think about caring for parents*". The experiences of many commenters were very different: they started caring for their parents as young adults. Commenters described how outsiders did not show any interest or understanding, neither did they acknowledge the commenters' family situation to be peculiar or potentially burdensome. Moreover, many commenters felt they were not allowed to talk about their caregiver role. One commenter realised they "*never really talked in depth to anyone about the complexities and regular worries they have felt having an older father*". One commenter stated that speaking out about it would put them at risk of "*being branded an ungrateful, unloving wretch*". Others also said they refused to speak out about it in order to protect their and their parents' dignity:

> There's this decades-long gentle decline, where the parent is only sort of ill, and in order to maintain their dignity (and yours), you have to do the caring in an invisible sort of way.

> To many people I am simply a man in his late-30s who still lives at home. Few people resist the chance to judge me harshly on that, and if I tell them that I am a caregiver, contempt frequently turns to pity—I am not sure which is worse!

**Emotional costs of caregiving.** Commenters who wrote about being a caregiver for their parent(s) when they were minors mentioned being burdened as well as feelings of loneliness ("*I have felt utterly alone at times*"), a sense of loss regarding the carefree childhood and the (greater) parental support they could have had. A few commenters did not talk about their own role as caregiver but recalled their older siblings assuming such a role as a minor. They remembered their siblings to attach negative feelings to their caregiver role, such as resentment and shock: "*My brother, 7 years older than me, bore some of the brunt of physical care and I know was deeply shocked.*"

Those who stated to have been a caregiver during adulthood mentioned fear and uncertainty towards their future and–similar to the experiences of the minors–they felt burdened having to take care of their parent(s). Moreover, they emphasised feeling "*out of step*" with their peers and with what was considered to be the normal life course:

> Feeling so out of step with my peers I felt it was abnormal, not to be spoken of.

> When I was in my early-20s I felt like a total failure because I hadn't achieved the degree or early-career milestones that my contemporaries had.

In line with this, most commenters who wrote about being a caregiver for their parent(s) felt compelled to remain silent. They hardly ever shared their experiences, concerns and troubles with others and their care work continued behind closed doors. A small number of commenters, however, presented their caregiving experiences in a more positive light. They attached a feeling of satisfaction to their caregiver role and to the influence it has had on their life and on the relationship with their parent(s):

> My dad died in his mid-70s, and I've moved in with my mum (who has a heart condition). The relationship I have with her now is one of the most rewarding I've experienced. I make sure she goes to her hospital appointments, we go out on trips (shopping or to her favourite gardens). I want to make sure we have some quality time together no matter how long we have left together.

There is the issue of 'generation squeeze' which happens to children born to older parents, who then have children late themselves: like me with an elderly dad in his early-80s, and a little son, who just started school. I deal with both ends of the life span—which keeps me busy—but is also rather wonderful.

## Discussion

The comments we studied displayed a wide variety of experiences with AAPs ranging from exclusively positive to extremely negative. Commenters often cited their parents' life experience, availability and commitment to parenting as important advantages of being raised by AAPs, which was consistent with the journalistic reports of Morris [21] and Yarrow [22]. Not all commenters, however, attributed their positive upbringing experiences to their parents' older age but instead, attributed their salutary upbringing to their parents' personal characteristics and qualities. Such comments reflect the idea that age in itself has little to do with someone's competence to raise a child. More negatively articulated childhood memories related to feeling different from peers and feeling embarrassed about having AAPs. Nevertheless, some commenters considered their childhood confrontation with social judgements about their parents as becoming increasingly irrelevant as they grew older, and one commenter even described how it taught them to disvalue the opinions of others. These findings show great resemblance with those of Morris [21] and Yarrow [22], suggesting that the challenges and/or benefits experienced and identified by (adult) children born to AAPs have not necessarily changed over the past few decades. Noteworthy in our dataset, however, is that a considerable number of commenters specifically (and exclusively) addressed the hardship of caring for their older parents. Commenters described their caregiving experiences in great detail, often using highly emotional and descriptive language, demonstrating a need to be heard and understood.

The majority of the negative experiences with AAP were attributed to the burdensome caregiver responsibilities associated with caring for older parents. The parents of these particular commenters became ill or disabled when the commenters were minors or young adults. Most commenters framed their caretaker role as a matter of filial responsibility, a self-evident duty they were under an obligation to fulfil. In western societies, however, the norm to provide care for parents is embedded in the expectation that such caregiving takes place during middle adulthood, usually defined as a life stage lasting from about age 40 to 65 [38]. Being a caregiver during early adulthood or even before, as a minor, is not expected. Therefore, society is maladjusted to the everyday life of these caregivers. Hence, we could argue that this assumed filial duty to provide care for one's elderly parents has become anything but obvious in current western societies which strongly endorse independence and individual responsibility.

As illustrated in the results, the commenters mentioned undertaking a range of caregiving activities, including both activities of daily living as well as providing emotional support and keeping the parents company. Interestingly, however, previous research has shown that parents' and children's ideas and attitudes towards caregiving within the family have changed over the last decades. Both parties do not expect adult children to take on personal care activities [39, 40]. Hence, the filial care duty nowadays is seen to primarily consist of providing emotional support and companionship for the parents [41]. Based on such a contemporary interpretation of the filial care duty, it seems plausible that many of the commenters in our dataset might have performed care activities which exceed what can be morally expected of them. However, given the lack of normalisation of young caregivers and a lack of accessible and affordable care services, it is likely that young caregivers, like the ones in our dataset, continue their care work alone and behind closed doors.

Only a few commenters shared their past experiences of caring for an older parent while being underage themselves. In the descriptions of these experiences we found multiple indications of what is regularly referred to as 'parentification'. Parentification as defined by Boszormenyi-Nagy and Spark [42] is "the distortion or lack of boundaries between and among family subsystems, such that children take on the roles and responsibilities usually reserved for adults". Our results reflect instances of both instrumental parentification (e.g., monitoring medical adherence) and emotional parentification (e.g., confiding the child with 'adult information', like money problems) [43]. Although we outlined above that providing emotional support might be at the core of a more contemporary interpretation of the filial care duty, research suggests that it is precisely this type of care activity that is potentially the most harmful when provided by minors [44]. However, consideration of the 'bigger picture' seems desirable: potential positive outcomes for the parentified child have been identified over the last few decades as well [45]. Hendricks and colleagues [45] point to the "multidimensional nature of parentification", thus urging us to assess a multitude of potentially relevant factors influencing the parentified child's experiences. The minor's perceived fairness of parentification [46] or their perceived control over their own behaviour within the family [47] have indeed been associated with more positive experiences of parentification. Therefore, we ought to be careful not to "overpathologise" [48] individuals who are or have been parentified during childhood and instead, consider their individual experiences and needs.

It must be noted that experiences of caring for parents are not exclusive to individuals born to older parents, and based on our data, it is impossible to judge whether the parents' health conditions were indeed related to their older age, or whether these were more exceptional cases in which the parents developed health issues relatively early (i.e., earlier than statistically expected). Similar caregiving experiences are found in children with a parent suffering from conditions like HIV/AIDS [49] or early onset dementia [50]. However, it remains a credible concern that reproducing at an older age increases the likelihood that a child will need to adopt the role of caregiver to their parent before they themselves enter middle age. Unlike conditions like dementia, which are often the result of an unlucky draw in the genetic lottery, prospective parents have a great degree of control over their reproductive decisions, raising further questions about parental responsibilities, and the age at which individuals should transition into parenthood.

## Strengths and limitations

Our study has several limitations. First, our data and study method do not allow for generalisations. Second, these data offer no background information on the commenters and no opportunity to ask follow-up questions [51] or verify whether the comments were an accurate reflection of the commenters' lived experiences. Our data could contain instances of erroneous or exaggerated representations of reality. Furthermore, stressing and even overstressing the positive side of a particular (life) story might be a result of people's tendency to eulogise events that happened in the past or to glorify deceased individuals [52]. Our study design and method also have important strengths. Research has found people to behave more honest and give unfiltered accounts online than in face-to-face set-ups [53]. Moreover, the commenters themselves volunteered their experiences and thoughts unsolicited and in a non-research setting as what they deemed relevant to share in the online discussion.

As a contribution to the literature on the lived experiences of (young) adults born to AAPs, our data exposed a pressing need for more societal understanding of what it means to be a caregiver for one's parents at a relatively young age. More understanding and normalisation could enable these young caregivers to dare to speak out about their family situation (more

often) and to share their experiences, concerns, and potential troubles with others. Caregiving experiences of (young) adults with AAPs should be further examined in terms of their frequency, the age at which a child or (young) adult starts to provide such care, and the impact this has on their wellbeing. In particular, future research needs to address the isolation of minors or young adults who take up caregiver roles for their AAPs.

Our findings call for an increased sensitivity of social policy to anticipate people's informal care activities. Considering the diversity of people's life courses and family circumstances, younger and older individuals should be treated alike and be offered the same flexible work arrangements, for example. Moreover, the education system and social services should be organised in such a manner as to facilitate the identification of young caregivers. These adolescents and young adults should be offered accessible and affordable support, while at the same time protecting them against unjustifiable social stigmas. Providing a safe space (whether it be an online platform or a physical place) where they can seek support and talk openly about their caregiving experiences can significantly contribute to their wellbeing, as it can reduce feelings of isolation and more generally, create a sense of community that can lift the burdens placed on individuals by the highly individuated, and self-reliant norms that predominate in western societies.

## Supporting information

**S1 Dataset. Comments from (adult) children born to older parents as shared to newspaper articles in The Guardian and The Daily Mail Online.**
(PDF)

## Acknowledgments

Special thanks to Jesse Gray for his valuable comments on this paper.

## Author Contributions

**Conceptualization:** Priya Satalkar-Götz, Veerle Provoost.

**Investigation:** Kato Verghote.

**Methodology:** Kato Verghote.

**Project administration:** Kato Verghote.

**Supervision:** Priya Satalkar-Götz, Guido Pennings, Veerle Provoost.

**Visualization:** Kato Verghote.

**Writing – original draft:** Kato Verghote.

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
