## [Decision Letter · Decision Letter 0]

24 Aug 2023

PONE-D-23-07136An invisible caregiver for visibly older parents: Experiences of (young) adults shared as comments to newspaper articles on advanced age parenthoodPLOS ONE

Dear Dr. Verghote,

Thank you for submitting your manuscript to PLOS ONE. After careful consideration, we feel that it has merit but does not fully meet PLOS ONE’s publication criteria as it currently stands. Therefore, we invite you to submit a revised version of the manuscript that addresses the points raised during the review process.

Consider the reviewers' comments and suggestions, in particular: (1) writing the article by a native English speaker; (2) clarifying the methodological aspects taking into account the reviewers' suggestions; (3) making adjustments to the discussion of the results, ensuring greater rigour and clarity.

We look forward to receiving your revised manuscript.

Kind regards,

Carla Maria Gomes Marques de Faria, Ph.D.

Academic Editor

PLOS ONE

Journal Requirements:

2.Please expand the acronym “ FWO” (as indicated in your financial disclosure) so that it states the name of your funders in full.

3.In your Data Availability statement, you have not specified where the minimal data set underlying the results described in your manuscript can be found. PLOS defines a study's minimal data set as the underlying data used to reach the conclusions drawn in the manuscript and any additional data required to replicate the reported study findings in their entirety. All PLOS journals require that the minimal data set be made fully available. For more information about our data policy, please see http://journals.plos.org/plosone/s/data-availability.

Reviewers' comments:

Reviewer's Responses to Questions

**Comments to the Author**

1. Is the manuscript technically sound, and do the data support the conclusions?

Reviewer #1: Yes

Reviewer #2: Yes

2. Has the statistical analysis been performed appropriately and rigorously? 

Reviewer #1: N/A

Reviewer #2: N/A

3. Have the authors made all data underlying the findings in their manuscript fully available?

Reviewer #1: Yes

Reviewer #2: Yes

4. Is the manuscript presented in an intelligible fashion and written in standard English?

Reviewer #1: No

Reviewer #2: Yes

5. Review Comments to the Author

Reviewer #1: I enjoyed reading this article, due to the relevance and topicality of the theme.

In fact, parenting is taking place at an increasingly advanced age and it is important to know the experience of the children of these aging parents, so that their specific needs can be met.

It is a topical, relevant and still little explored subject, which makes this study valuable.

Overall, this paper may be of interest to the PLOS ONE readership, however major changes are necessary for it to be published, which essentially have to do with its writing.

Thus, the first condition to increase the probability of being published is to be written, in full, by someone who is native or who has a good expertise of the English language. In fact, the article presents a confusing writing that does not respect, at a morphological and syntactic level, the basic rules of the English language. This impairs the understanding of the study’s content. I present some examples of sentences that fail or are difficult to understand, although this issue is transversal to the entire article:

- The few empirical studies available show similar or more positive child outcomes compared to children born to younger parents

- feared source of such poor psychosocial outcomes is a suggested lack adequate parenting skills of older parents who would thus not be able to fulfil their parental duties

- The connected concern is that since the children will be younger when their parents start experiencing more serious health problems

- However, where emerging adulthood is seen as a time in which a gradual process of role-reversal and confrontation with filial duty begins this may be a more sudden experience taking place at na earlier age for people born to AAPs. Overall, mainly quantitative studies point to similar or even more positive child outcomes compared to children born to younger mothers

- TG and DMO yielded the most articles written on the subject of older parenthood

- DMO has said to be right and targets lower-middle-class readers.

Introduction

- The introduction presents a good framework about late parenthood.

- As you refer in the paper, there is a lack of consensus regarding what is considered late parenthood. However, it is important to present what has been considered as such, that is, to present age references, both for the first child and for subsequent ones – What has been considered late parenthood in terms of age, in the literature?

- At the end of the introduction, you refer several times to “popular press” – it is important to clarify what popular press is

- At the end of the introduction, you also refer “By studying data stemming from stakeholders expressing in an unsolicited way what they consider relevant for the debate on AAPd we hope to give voice to these stakeholders and initiate further research” (lines 90-92) – Here, I suggest that you analyze whether the “participants” are really stakeholders. Stakeholders are key informants, and, in this case, it seems that they are simply people who volunteer to answer and not people intentionally selected to give their opinion on a concrete subject, in which they have expertise and knowledge.

Methodology

- In the methodology, you once again refer to the issue of stakeholders – “We chose to limit the search to newspaper articles that presented the perspective of stakeholders because they described this topic from a personal perspective thereby indirectly eliciting commenters’ own experience”. – It is necessary to clarify who are considered stakeholders in this study and what were the criteria for being considered as such.

- It is also referred: “Of all accessible comments (1.248), we created a data subset of comments in which it was clear that the commenter was an individual born to (an) older parent(s). Rather than a precisely defined definition of ‘older parenthood’ by taking a particular age cut-off as a benchmark, we used the commenters’ descriptions and experiences of having or living in a family with older parents”. – Here I pose the following questions: How did it become clear that the commentator was a child of elderly parents? What were the criteria for being considered as such? Also, one more language issue: it is not correct to say “defined definition”.

- It is necessary to clarify the meaning of this statement because it is not clear: “The unit of our analysis was comments; therefore, we only made claims about comments and not about the commenters”.

Data Analysis

- In data analysis procedures, you should not name the authors of the article for each task performed.

- It is necessary to clarify the meaning of this statement: “Additional auditing cycles were organised in collaboration with all co-authors”

Results

In general, the results are well structured, presenting the 3 main themes found and some categories within the themes.

I suggest that, for a better orientation of the reader, you refer right from the start, in each theme, which were the categories (and sub-categories) defined.

- It is necessary to clarify the meaning and purpose of this statement: “Most of the comments formulated by individuals who shared their personal experiences as children of advanced age parents (AAPs) were formulated as both a reaction to the respective newspaper article and a contribution to the debate on advanced age parenthood (AAPd)”

- You refer: “Overall, there was a comparable number of comments taking a positive and a negative stance towards AAPd.” – What do you mean by comparable? How did you come to that conclusion? Did you count? In what way?

- When themes are presented (between lines 175 and 181), it is important to put the name of the theme, between parentheses or hyphens.

- Between lines 210 and 218 several ideas/contents are presented that are not categorized. It is important to create categories/sub-categories for them. These contents are too different from the “Not missing out” category to be integrated into it.

- I suggest that you also create categories/sub-categories within the second and third themes - Being a Child of Visibly Older Parents and Doing the Caring in an Invisible Sort of Way - as they present a lot of rich and different information that can be grouped together and better understood if categories are created to organize it.

Discussion

In the discussion, a fairly complete integration of the results with previous research is made.

Reviewer #2: Dear authors

I was very pleased to review the manuscript entitled: An invisible caregiver for visibly older parents: Experiences of (young) adults shared as comments to newspaper articles on advanced age parenthood.

This qualitative study focuses on the experiences presented by (young) adults who identified as born to older parents offered in response to a selection of newspaper articles on the topic.

The subject addressed in this research is relevant, and its importance is likely to increase in the coming years, as the average age of mothers and fathers at first birth has increased in recent decades.

Overall, the article is well written, and the qualitative analysis procedures are adequated and well documented, thus allowing replication of the analyses. Although the literature review is well conducted and the results are robust, in some cases, the authors could better support the results of the cited studies. Concerning the discussion section, this could be more developed.

In order to improve the quality of the manuscript I suggest the revision of the following points:

- In the Introduction section (lines 80-82) the authors stated: "The only available qualitative accounts of (adult) children born to older parents date from the last century and are more journalistic in nature..." Since these are the only qualitative studies regarding (adult) children born to older parents, a deep analysis of these studies' results is needed.

- Also in the Introduction section, the authors should indicate more clearly what are the main contributions to the literature of this research.

- In the Method section, the authors should indicate why they used reflexive inductive thematic analysis and the benefits of using this approach. Besides, since the audience of the analyzed newspapers is global, meaning readers are from several countries, the authors should state this.

- In the Discussion section, the authors focused mainly on the third theme: the invisibility of taking up an untimely caregiver role for AAPs. An in-depth discussion of the remaining two themes would be necessary to highlight the study's results correctly.

Regarding the third theme, the authors did not present/discuss the main differences between young adult children caregivers and older adult children caregivers regarding burdensome caregiver responsibilities that adversely affect their lives. Are these two groups so different? Additionally, the authors should highlight the study's practical implications and state some public social and health policies to support these younger caregivers.

Finally, I want to reinforce the quality of the study and encourage the authors to continue to explore this topic in future studies.

6. PLOS authors have the option to publish the peer review history of their article (what does this mean?). If published, this will include your full peer review and any attached files.

Reviewer #1: No

Reviewer #2: **Yes: **Fátima Cristina Senra Barbosa

---

## [Author Response · Author response to Decision Letter 0]

9 Oct 2023

Dear Editor,

Dear reviewers,

Thank you for this opportunity to improve our manuscript and to make it ready for publication in PLOS ONE. We think the reviewers brought up some excellent points which we have tried to address as good as possible. Please find all information about the requested adjustments in our Response to Reviewers file. We have also made adjustments to the manuscript, cover letter and supporting information file as to address the additional journal requirements the Editor referred to. 

Thank you again for your consideration of this manuscript.

Sincerely,

Kato Verghote

---

## [Decision Letter · Decision Letter 1]

14 Nov 2023

An invisible caregiver for visibly older parents: Experiences of (young) adults shared as comments to newspaper articles on advanced age parenthood

PONE-D-23-07136R1

Dear Dr. Verghote,

We’re pleased to inform you that your manuscript has been judged scientifically suitable for publication and will be formally accepted for publication once it meets all outstanding technical requirements.

Kind regards,

Carla Maria Gomes Marques de Faria, Ph.D.

Academic Editor

PLOS ONE

Additional Editor Comments (optional):

Reviewers' comments:

Reviewer's Responses to Questions

**Comments to the Author**

1. If the authors have adequately addressed your comments raised in a previous round of review and you feel that this manuscript is now acceptable for publication, you may indicate that here to bypass the “Comments to the Author” section, enter your conflict of interest statement in the “Confidential to Editor” section, and submit your "Accept" recommendation.

Reviewer #1: All comments have been addressed

Reviewer #2: All comments have been addressed

2. Is the manuscript technically sound, and do the data support the conclusions?

Reviewer #1: Yes

Reviewer #2: Yes

3. Has the statistical analysis been performed appropriately and rigorously? 

Reviewer #1: N/A

Reviewer #2: N/A

4. Have the authors made all data underlying the findings in their manuscript fully available?

Reviewer #1: Yes

Reviewer #2: Yes

5. Is the manuscript presented in an intelligible fashion and written in standard English?

Reviewer #1: Yes

Reviewer #2: Yes

6. Review Comments to the Author

Reviewer #1: (No Response)

Reviewer #2: Dear authors

Thank you very much for the responses and the changes made to the manuscript.

Best regards

7. PLOS authors have the option to publish the peer review history of their article (what does this mean?). If published, this will include your full peer review and any attached files.

Reviewer #1: No

Reviewer #2: **Yes: **Fátima Cristina Senra Barbosa

---

## [Editor Report · Acceptance letter]

20 Nov 2023

PONE-D-23-07136R1 

An invisible caregiver for visibly older parents: Experiences of (young) adults shared as comments to newspaper articles on advanced age parenthood 

Dear Dr. Verghote:

I'm pleased to inform you that your manuscript has been deemed suitable for publication in PLOS ONE. Congratulations! Your manuscript is now with our production department. 

Kind regards, 

on behalf of

Professor Carla Maria Gomes Marques de Faria 

Academic Editor

PLOS ONE